# Genome-Wide Association Studies (GWAS) Approaches for the Detection of Genetic Variants Associated with Antibiotic Resistance: A Systematic Review

**DOI:** 10.3390/microorganisms11122866

**Published:** 2023-11-27

**Authors:** Jeanneth Mosquera-Rendón, Claudia Ximena Moreno-Herrera, Jaime Robledo, Uriel Hurtado-Páez

**Affiliations:** 1Bacteriology and Mycobacteria Unit, Corporation for Biological Research (CIB), Medellín 050034, Colombia; jmosquerar@unal.edu.co (J.M.-R.); jrobledo@cib.org.co (J.R.); 2Microbiodiversity and Bioprospecting Group (Microbiop), Department of Biosciences, Faculty of Sciences, Universidad Nacional de Colombia, Medellín 050034, Colombia; cxmoreno@unal.edu.co

**Keywords:** genome-wide association study, antimicrobial resistance, genetic variants, bacteria

## Abstract

Antibiotic resistance is a significant threat to public health worldwide. Genome-wide association studies (GWAS) have emerged as a powerful tool to identify genetic variants associated with this antibiotic resistance. By analyzing large datasets of bacterial genomes, GWAS can provide valuable insights into the resistance mechanisms and facilitate the discovery of new drug targets. The present study aimed to undertake a systematic review of different GWAS approaches used for detecting genetic variants associated with antibiotic resistance. We comprehensively searched the PubMed and Scopus databases to identify relevant studies published from 2013 to February 2023. A total of 40 studies met our inclusion criteria. These studies explored a wide range of bacterial species, antibiotics, and study designs. Notably, most of the studies were centered around human pathogens such as *Mycobacterium tuberculosis*, *Escherichia coli*, *Neisseria gonorrhoeae*, and *Staphylococcus aureus*. The review seeks to explore the several GWAS approaches utilized to investigate the genetic mechanisms associated with antibiotic resistance. Furthermore, it examines the contributions of GWAS approaches in identifying resistance-associated genetic variants through binary and continuous phenotypes. Overall, GWAS holds great potential to enhance our understanding of bacterial resistance and improve strategies to combat infectious diseases.

## 1. Introduction

Antimicrobial resistance (AMR) has emerged as a critical public health threat in the 21st century. According to estimates from 2014, the total number of deaths due to AMR is projected to increase dramatically from 700,000 per year to 10 million per year by 2050, with a potential cost to the world of up to 100 trillion USD [1]. This alarming trend is compounded by the fact that the development and dissemination of AMR compromise the effective treatment of infectious diseases, underscoring the urgent need for comprehensive action to address AMR’s proliferation [2].

In 2021, the World Health Organization (WHO) reported that pathogenic bacteria such as *Klebsiella pneumoniae*, *Staphylococcus aureus*, *Pseudomonas aeruginosa*, *Neisseria gonorrhoeae*, and *Escherichia coli*, causing hospital-acquired infections and common bacterial diseases, such as urinary tract infections, sepsis, sexually transmitted infections, and specific forms of diarrhea, have shown alarming levels of resistance to widely used antibiotics on a global scale [3]. Furthermore, the challenge intensifies with the emergence and spread of antibiotic-resistant strains of *Mycobacterium tuberculosis*, which significantly hampers efforts to contain the worldwide tuberculosis epidemic [3].

A comprehensive understanding of the genetic bases of bacterial resistance holds the potential to decipher the biological mechanisms driving infectious diseases. This knowledge would significantly bolster the capacity to develop novel medications, vaccines, and predictive tools while enhancing disease surveillance and overall public health [4]. The advancement of sequencing technologies, coupled with the extensive growth of bacterial genomic datasets and phenotypic data, have opened up new opportunities for large-scale computational approaches, such as bacterial Genome-Wide Association Studies (GWAS) [5,6,7,8]. This field allows an understanding of the role of genetic variants such as Single-Nucleotide Polymorphisms (SNPs), insertion and deletion variants (indels), and the presence or absence of genes that influence pathogen phenotypes associated with drug resistance, virulence, host specificity [9], high and low prevalence sublineages [10] among others.

The primary purpose of GWAS is to identify statistically significant associations that may indicate causal relationships between genotype and phenotype while eliminating spurious associations arising from confounding factors [4]. Consequently, there is an important interest in implementing GWAS approaches in the study of pathogenic bacteria to advance our understanding of infectious disease risks and identify genetic variants driving bacterial resistance [5,6,7,8] while offering immense potential for development and improvements in disease management and treatment.

Important works have explored the adaptation of GWAS approaches used in human genetics to the realm of bacterial genetics, considering the substantial genomic differences (genetic recombination, linkage disequilibrium, population structure, and genetic variants, among others) between both [5,6,7]. This has generated the application and development of diverse GWAS approaches to overcome bacterial genomic challenges [4,8,11,12,13]. The aim of this systematic review is to describe bacterial GWAS approaches employed to identify genetic variants associated with antimicrobial resistance, as well as alternative strategies for discerning genotype-phenotype associations in bacteria.

## 2. Materials and Methods

This review was designed according to the PRISMA guideline for publishing systematic review checklists (Appendix A).

### 2.1. Eligibility Criteria

The eligibility criteria for inclusion in the review were based on the PICOS (Population, Intervention, Comparison, Outcome, and Study design) framework: (1) Population will be bacterial species that are known to cause infections in humans or animals. (2) Interventions involved GWAS for the identification of bacterial resistance determinants. (3) Comparison groups were non-GWAS studies that have investigated bacterial resistance determinants. (4) Outcomes were the genetic variants associated with bacterial resistance. (5) The study design encompassed genome-wide association studies that were published in peer-reviewed journals. Additionally, only publications in English were considered. Studies reporting on phenotypes other than antibiotic resistance were excluded. Conference abstracts, editorials, review articles, book chapters, and studies that applied GWAS approaches in organisms other than bacteria (viruses, fungi, humans) were also excluded. Full inclusion and exclusion criteria are provided in Table 1.

### 2.2. Information Sources and Search Strategy

The search strategy aimed to identify relevant studies that shed light on the mechanisms and genetic factors behind bacterial resistance to antibiotics. The search was conducted on major databases, such as PubMed and Scopus, which are widely used repositories of scientific literature. To ensure comprehensiveness, a mix of MeSH terms and keywords were employed, including “Genome-Wide Association Study”, “antibiotic resistance”, “antimicrobial resistance”, and “GWAS”. Appendix A provides a list of the search strategies tailored to the specific requirements of each database. Only articles published in the full text before 7 February 2023 were considered. The study was initiated on 17 August 2022.

### 2.3. Data Management

All citations and publications abstracts obtained from the results of the search equations in Scopus and PubMed were exported to Rayyan (https://rayyan.qcri.org), a web application designed for systematic review of articles. Using this tool, the entire selection process, including the review of titles and abstracts, elimination of duplicate articles, and selection of studies that meet the eligibility criteria, was managed.

### 2.4. Selection Process

A single reviewer conducted a two-phase review of the studies. The first phase involved an initial screening of the articles to evaluate the relevance of the title and abstract. During this phase, the reviewer identified those articles that meet the inclusion criteria. In the second phase, the reviewer thoroughly evaluated the full text of the articles that passed the initial screening to determine if they met the eligibility criteria. The same inclusion criteria were applied during both phases of the selection process to ensure consistency in the review and minimize bias. The reviewer also documented the reasons for excluding articles that did not meet the inclusion criteria.

### 2.5. Data Extraction

An Excel spreadsheet was employed to collect information from each GWAS study in the systematic review. This information played a crucial role in achieving the review’s objectives. The essential data collected included study characteristics, authors, year of study, study objectives, bacterial species studied, sample size, resistance phenotype, phenotypic traits, genetic variants associated with antibiotic resistance, GWAS software used, employed GWAS approach, population structure control methods (useful for analyzing the statistical methods employed in the studies), parameter statistics (multiple test corrections), identified significant genetic variants findings and validation of the results. This information is available in Appendix A.

## 3. Results

### 3.1. Study Selection

The systematic review aimed to analyze GWAS approaches used for detecting genetic variants related to the resistance phenotypic traits in bacteria. The PRISMA2020 flowchart (Figure 1) presents a visual representation of eligible studies identified through the bibliographic search. As shown in Figure 1, the search strategy was comprehensive and included the use of two major databases, PubMed and Scopus, resulting in a total of 1347 studies initially identified. After removing duplicate studies, 601 studies remained for screening based on their titles and abstracts. During the screening process, studies that did not perform GWAS were excluded, resulting in the removal of 268 studies. Furthermore, studies that conducted GWAS in non-bacterial species (186 studies) or in the incorrect phenotype (50 studies) were also excluded. In addition, studies that did comply with the publication type were excluded, which included review articles, case reports, and conference abstracts (37 studies). After the initial screening process, 57 articles were left for full-text evaluation. During the full-text evaluation, articles that did not meet the inclusion criteria were excluded, resulting in the removal of 17 studies. Out of the seventeen excluded studies, eight did not use GWAS, and nine used a phenotype other than antibiotic resistance. Finally, the systematic review identified a total of 40 studies that met the inclusion criteria (Table 2).

### 3.2. Study Characteristics

In this review, we have compiled 33 studies published in the last decade that utilized GWAS to identify genetic variants associated with resistance to various antimicrobials, as well as seven publications related to tools developed for GWAS studies. Among the studies included in this review, 15 focused on human pathogenic bacteria, including *Mycobacterium tuberculosis* (the causative agent of tuberculosis), *Escherichia coli* (causes a range of diseases both within and outside the intestinal tract), *Staphylococcus aureus* (which causes skin infections, respiratory infections such as sinusitis and pneumonia, food poisoning and sepsis), *Neisseria gonorrhoeae* (which causes sexually transmitted diseases as urethritis, cervicitis, and neonatal infections among others), *Streptococcus pneumoniae* (a major cause of pneumonia), *Neisseria meningitidis* (which causes meningitis), as well as *Acinetobacter baumannii*, *Pseudomonas aeruginosa*, and *Klebsiella pneumoniae* (responsible of hospital-acquired infection) among others. Advancements in whole-genome sequencing technologies have significantly contributed to the increased availability of bacterial genomic datasets. 

### 3.3. Application of GWAS for the Identification of Antimicrobial Resistance Loci

Recent advancements have enabled the application of bacterial GWAS for the identification of genetic variants associated with antimicrobial resistance. Bacterial GWAS studies have provided valuable insights into the genetic basis of resistance in various bacteria, deepening our understanding of underlying mechanisms such as the acquisition of resistance genes, mutations in essential genes, and alterations in regulatory elements. Additionally, these studies have contributed to identifying novel resistance determinants and characterizing resistance mechanisms, which can inform the development of new therapeutic strategies.

The studies included in this review showed that bacterial GWAS analyses aimed at identifying genetic variants associated with antimicrobial resistance involve multiple stages (Figure 2). Firstly, bacterial isolates are sampled from various sources, including clinical environments, animal reservoirs, or environmental samples. The inclusion of isolates representing different conditions of the resistance phenotype of interest allows for a comprehensive analysis of genetic variants associated with resistance. Subsequently, the antimicrobial resistance phenotype of each isolate is measured, typically through drug susceptibility tests such as minimum inhibitory concentration (MIC) assays or disk diffusion tests. This enables the classification of isolates as susceptible or resistant to specific antimicrobial agents, following the EUCAST or CLSI breakpoint standards. Concurrently, the genomes of bacterial isolates are sequenced using whole-genome sequencing technologies, providing genomic information for each included isolate.

Next, the sequencing data are analyzed using specialized bioinformatics algorithms and tools to identify genetic variants, such as SNPs [4,8,12,15,16,19,20,21,22,23,24,26,27,28,30,31,34,36,37,39,41,42,45,49,50], indels [22,24,28,41,49] as well as the presence or absence of genes [4,12,14,16,17,20,21,24,25,28,30,36,42,45,47]. The identified genetic variants are then subjected to statistical analysis to determine their association with the resistance phenotype. Various statistical approaches can be employed, including Chi-square tests, Fisher’s exact test [22], Cochran–Mantel–Haenszel Test [21,44], logistic regression [17], linear mixed models [19,27,28,29,31,32,39,46,47,48,49], approaches based on phylogenetic [4,15,19,23,24,26,28,30,34,36,47], or a mixed approaches [25,31,37]. Furthermore, it is relevant to consider additional factors to mitigate the risk of obtaining false-positive outcomes and minimize the potential for the Family-wise error rate (FWER) when applying corrections for multiple testing. Methods such as Bonferroni’s correction [4,12,15,21,22,24,25,27,29,30,32,33,35,36,38,40,41,44,46,47,48,49] and the FDR-Benjamini–Hochberg correction [8,18,20,23,26,28,31,34,39,42,43,50] might be necessary to address the potential inflation of false-positive results when evaluating numerous genetic markers.

Finally, in some studies, the findings are validated using independent cohorts of bacterial isolates or by conducting genetic tests, molecular assays, or bioinformatic analyses to further investigate the functional implications of the variants identified.

### 3.4. Bacterial Genome-Wide Association Studies (GWAS) Approaches

In this article selection, we have identified a diversity of GWAS approaches used to identify genetic variants associated with antimicrobial resistance. Our review revealed that 16 studies focused on non-phylogenetic methods, including the Fisher exact test, Cochran–Mantel–Haenszel (CMH) test, logistic regression model, and linear mixed model. In contrast, five studies chose phylogenetic approaches. Furthermore, five articles chose the alignment-free *k*-mers-based approach for their research. In parallel, another five studies employed a mixed approach that integrated various methods into a single analysis. Additionally, it is worth noting that six studies combined non-phylogenetic and phylogenetic approaches, and one study stood out by integrating three distinct methods: non-phylogenetic, *k*-mers-based without alignment, and a mixed approach, thereby leveraging the strengths of each in the study.

#### 3.4.1. Non-Phylogenetic Approach

In this approach, a variety of statistical tests and models had been employed. Initially, statistical tests based on the contingency table were utilized, including the Chi-Square Test, Fisher’s exact test [22], and Cochran–Mantel–Haenszel Test [21,44] for the categorical traits. In the study conducted by Mobegi et al. [21], the association tests were performed using the Cochran–Mantel–Haenszel Test, and the delimitation of groups of strains was employed to correct for genetic relationships. This approach aimed to identify SNPs and/or genes associated with resistance to various antibiotics, including penicillin, trimethoprim, cotrimoxazole, fluoroquinolones, and macrolides, in four genetically distinct populations of *S. pneumoniae*. Notably, specific SNPs associated with trimethoprim and cotrimoxazole resistance were found within genes responsible for folate metabolism, including *dyr*, *folE*, and *folP*.

For binary traits, logistic regression models are employed to evaluate the association between the presence or absence of a genetic variant, such as SNPs, and the binary result (0 = Sensitive or 1 = Resistant) based on standard concentrations for DST or cutoffs for MICs [17]. For quantitative traits, linear regression models are used, where linear model coefficients (LM) reflect the statistical correlations between the presence of an allele and the appearance of a specific phenotype. These models are adjusted for each SNP combination and quantitative trait, such as the log of MICs [31].

Different approaches have been developed to reduce the impact of population structure on bacterial GWAS, such as Linear Mixed Models (LMM) and dimensionality reduction techniques like Principal Components Analysis (PCA) and multidimensional scaling. One common approach is the use of LMM, which incorporates fixed and/or random effects as predictors (covariates) to account for population structure and kinship relationships. These models include a kinship matrix or genetic similarity matrix (GSM) estimated from SNPs or other genetic variants. Incorporating the kinship matrix as a covariate (random effects) in the LMM can help identify locus effects more accurately by reducing the potential for false positive associations among highly related or genetically related isolates [28,31,46,49].

In the study conducted by Fathat et al. [28], the GWAS method based on the Linear Mixed Model, specifically GEMMA, was employed, and population structure was incorporated by computing a Genetic Relatedness Matrix (GRM). This matrix measured genetic similarity as the covariance between the genetic variant vectors of 1526 *M. tuberculosis* isolates. The study identified 13 non-canonical loci that were associated with resistance to different anti-tubercular agents.

In dimensionality reduction-based approaches, a kinship matrix is employed and projected onto a smaller number of components. Subsequently, a specific number of these components are selected as fixed effects factors in the LMM or logistic regression for binary phenotypes to incorporate lineage-specific effects and control for population structure [51]. In the bacterial GWAS focusing on antibiotic resistance phenotype, two commonly used methods are observed. The first method involves generating a kinship matrix from PCA and using it as fixed effects in an LMM [12,19,24,50]. In this approach, the regression coefficients calculated for principal components (PC) represent the variations in phenotypes across lineages. Each PC is examined to assess its impact on the phenotype. Defining lineages based on PC, rather than phylogenetic branches or genetic clusters, helps minimize the loss of power to detect lineage-level associations caused by correlations between lineages [12].

In 2016, a bacterial GWAS tool was developed to control for population structure by exploiting the connection between PC and LMMs [12]. This tool decomposes the random effects estimated by the LMM to obtain coefficients and standard errors for each PC. A Wald test was then used to evaluate the significance of the association between each lineage and the phenotype [12]. *Bugwas* has demonstrated its effectiveness in detecting genes and genetic variants associated with antimicrobial resistance in different bacteria strains, including *A. baumanni* [18], *M. tuberculosis*, *S.aureus* [8,12], *P. aeruginosa* [8], *M. abscessus* [49], *E. coli*, *K. pneumoniae* [12].

One notable study, published in 2018, conducted a GWAS to identify genetic variants associated with resistance to 14 anti-TB drugs using a large dataset of 6465 *M. tuberculosis* clinical isolates from more than 30 countries. This study employed an LMM that included principal components for the main *M. tuberculosis* lineage, sublineage effects as fixed effects, and an SNP-inferred kinship matrix as a random effect. These approaches were utilized to account for highly related samples and fine-scale population structures that could potentially result from outbreaks [24].

The second method entails employing the multidimensional scale of genetic similarities, based on a pairwise distance matrix, as a covariate in a linear regression model [33,37]. In a study conducted by McDougall et al. [33], a GWAS was undertaken using the oxacillin MICs on 265 clinical isolates to determine the genetic basis of the increase in MIC for β-lactams amongst *Streptococcus uberis*. For this, each *k*-mers (i.e., DNA words of length *k*) is fitted to an LMM, and for the population structure, a distance matrix is constructed from a random subsample of the *k*-mers, on which metric multidimensional scaling (MDS) is performed. Furthermore, to facilitate LMM-based GWAS methods and control for population structure, tools such as *pyseer* have been developed. These tools offer the ability to construct the kinship matrix using genetic distances based on genetic variants [8,30,37,39,48] or extracting patristic distances from phylogeny [32,39,41] for applied LMM. These features enhance the accuracy and flexibility of population structure control in bacterial GWAS, allowing for more robust and comprehensive analyses of genetic associations.

#### 3.4.2. Alignment-Free *k*-mer Based Approach

Another proposed approach is the use of *k*-mer-based methods without alignment, which analyze the association between the presence or absence of short nucleotide fragments (*k*-mers) of 20–30 bp and bacterial phenotypes such as drug resistance [8,13,44]. This approach does not require a reference genome and allows analysis without the need to assemble the genomes of the bacterial isolates. The *k*-mers capture different genetic variants, including SNPs, indels, insertions/deletions located in coding or non-coding regions, and the presence or absence of genes, as well as large mobile genetic element insertions, without the need for prior identification or definition [8,44].

One tool that utilizes the previous approach is *pyseer*, which identifies *k*-mers and their association with antibiotic resistance phenotypes using LMM (Fast-LMM) for each *k*-mer. Additionally, multidimensional scaling of a pairwise distance matrix is performed, and these components are included as fixed effects in each regression to control for population structure [13]. This approach has been employed in various GWAS analyses in different bacteria, including *M. tuberculosis*, *S. aureus*, *P. aeruginosa* [8], *N. gonorrhoeae* [29,32,48], *A. baumannii* [30], aiming to discover and detect genetic variants associated with resistance to different antibiotics.

Recently, one of the largest GWAS studies associated with resistance phenotype in *M. tuberculosis* was published. In this study, a novel approach was developed to test associations in oligopeptides (11-mers) and oligonucleotides (31-mers) using LMM to detect relevant genetic variations in both coding and non-coding sequences. This approach avoids a reference-based mapping approach that may inadvertently overlook significant variations [46].

Also, tools such as DBGWAS have been developed. DBGWAS utilized compacted De Bruijn graphs (cDBG) to produce interpretable genetic variants associated with different phenotypes. This tool employs LMM and utilizes *bugwas* to test lineage effects. DBGWAS reduces the number of association tests by constructing *unitigs*, which are longer sequences obtained by overlapping *k*-mers [8]. This approach has demonstrated effectiveness in detecting genes and genetic variants associated with antimicrobial resistance in different bacterial strains, including *M. tuberculosis*, *P. aeruginosa* [8], *S. aureus* [8,42], *Staphylococcus capitis* [38,43], *Mycoplasma bovis* [40].

Additionally, CALDERA, a tool designed to facilitate bacterial GWAS, has recently been developed. CALDERA analyzes closed connected subgraphs (CCS) within the de Bruijn graph representing genomic *k*-mers. By treating polymorphic genes as a unified entity, CALDERA improves the power and interpretability of *k*-mers-based GWAS [44].

#### 3.4.3. Phylogenetic Approach 

A novel approach has emerged to address the population structure in bacteria, utilizing methods based on convergent evolution. This approach offers a distinct advantage over LMM by incorporating individual mutation events along a phylogenetic tree, enabling a comprehensive understanding of evolutionary relationships between strains [31]. A prominent tool that applies this approach is the Phylogenetic Convergence Test (PhyC) [19,24,28]. PhyC employs permutation testing to ascertain the significance of associations between observed nucleotide changes at specific sites (in inferred branches of a phylogenetic tree) and the acquisition of resistance (inferred in internal nodes using maximum parsimony). By calculating the probability that this observation occurs by chance, PhyC assesses the significance of the association [11].

Furthermore, a tool called treeWAS has been proposed, which is based on systematic statistical analysis and does not rely on prior hypotheses regarding potential associations at candidate loci. This method employs a phylogenetic tree as input to determine the genetic relatedness between strains. It conducts three independent association tests (terminal, simultaneous, and subsequent) for all loci, generating scores that detect associations with binary, categorical, or continuous phenotypes. To comprehensively explore potential signals of association, these tests are executed in parallel, and simulated datasets are utilized to evaluate spurious associations [4]. The authors recommend integrating ClonalFrameML [52] into treeWAS to construct the phylogenetic tree. By leveraging ClonalFrameML, more accurate phylogenetic trees can be generated, particularly when dealing with species exhibiting a high recombination rate [4]. 

In the study conducted by Collins and Didelot [4], the treeWAS tool was applied to perform a GWAS analysis on a dataset comprising 171 *N. meningitidis* isolates exhibiting penicillin resistance phenotypes. Both binary phenotypes (resistant and susceptible) and continuous phenotypes, by transformed minimum inhibitory concentration (MIC) values, were evaluated. The results revealed significant SNPs predominantly located in the NEIS1753 (*penA*) gene, as well as other relevant genes, providing valuable insights into the underlying genetic mechanisms of penicillin resistance in this bacterial species.

Furthermore, the applicability of treeWAS has been demonstrated in the identification of genes and genetic variants associated with antimicrobial resistance in various bacterial strains, such as *M. tuberculosis* [28], *A. baumannii* [30], *Corynebacterium diphtheriae* [36], and *Streptococcus pneumoniae* [47]. Several of these studies utilized treeWAS to compare results obtained with other tools based on LMM [28,30,47].

Another convergence phylogenetic-based method called phyOverlap was developed, inspired by the work of Farhat et al. [11]. This approach is based on reconstructing ancestral sequences using the maximum parsimony technique. This approach enables the comparison of current genetic sequences within each bacterium with their respective ancestral iterations. Subsequently, computational techniques, such as Fitch’s algorithm, are employed to ascertain the most probable states of the ancestral attributes located within the internal nodes of the phylogenetic tree. This step is complemented by the execution of statistical tests, aiming to scrutinize the relationship between genetic variations and drug resistance within a collection of bacterial isolates [23]. This method has been utilized in several publications to identify genetic variants that influence the susceptibility of *M. tuberculosis* populations to antibiotics, specifically isoniazid [23,34] and ethionamide [26].

#### 3.4.4. Mixed Approach

Antimicrobial drugs can display multiple resistance mechanisms, frequently marked by the presence of multiple resistance mutations within a single strain [31]. Therefore, in order to enhance our understanding of multiple drug resistance, it is crucial to not only identify the primary mutations that directly contribute to drug resistance but also to elucidate the secondary mutations that facilitate the gradual buildup of resistance [16]. This poses a significant challenge for conventional GWAS methods, as they encounter difficulties in detecting these resistance mechanisms due to their weaker statistical associations.

To address these challenges, the GWAMAR method was developed as a pipeline tool designed to identify mutations associated with drug resistance through comparative analysis of whole-genome sequences obtained from closely related bacterial strains. GWAMAR utilizes statistical scores, including the introduced tree-generalized hypergeometric score (TGH), which incorporates phylogenetic information and weighted support to account for the evolutionary pressures resulting from drug treatment and the geographical distribution of drug resistance mutations [16]. In a study conducted by Wozniak et al. [16] to evaluate the effectiveness of GWAMAR, two large datasets comprising 173 and 1398 samples, respectively, were collected from publicly available data on *M. tuberculosis*. This study successfully identified a set of novel putative mutations associated with drug resistance and additionally detected potential compensatory mutations.

A recently proposed approach called the Evolutionary Cluster-based Association Test (ECAT), utilizes an LMM that incorporates information on homoplasy, which refers to the occurrence of multiple independent mutations at the same site. By incorporating this information, ECAT improves the identification of genes associated with drug resistance [31]. ECAT, a method proposed by Lai et al. [31], involves a preprocessing step to identify hypervariable regions in the genome characterized by statistically significant clusters of evolutionary changes. Subsequently, an LMM is applied to test for associations within these identified regions, thus improving the detection of genes associated with drug resistance [31]. An evaluation of ECAT using a collection of over 600 multidrug-resistant (MDR) *M. tuberculosis* clinical isolates from Lima, Peru, demonstrated that ECAT outperformed an LMM approach, both site-based and gene-based, in detecting known resistance mutations to various anti-TB drugs, even mutations with lower prevalence and weaker associations [31].

Hungry, Hungry SNPos (HHS), an approach proposed by Libiseller-Egger et al. [37], utilizes a scoring heuristic to select an initial batch of alleles and employs an iterative process known as “cannibalism,” where selected variants are compared and evaluated, resulting in the removal of variants with lower scores or lesser significance. The primary objective of HHS is to address the confounding effects arising from co-occurring resistance markers. A study utilizing 3574 isolates of *M. tuberculosis* genomic dataset, focusing on the virulent Beijing strain type, included phenotypic resistance data for five drugs: isoniazid, rifampicin, ethambutol, pyrazinamide, and streptomycin, demonstrated the superior robustness of HHS compared to conventional methods. The study findings highlight HHS’s capability to detect resistance-associated variants that traditional methods would likely miss [37].

Similarly, Schubert et al. [27] introduced an innovative systematic GWAS aimed at investigating the landscape of antimicrobial sensitivity and resistance in *N. gonorrhoeae*. Notably, this study introduced a novel and widely applicable approach for performing epistatic GWAS in bacteria, a methodology with significant implications for deciphering genomic-scale epistatic interactions. At the core of this approach lies Evolutionary coupling analysis, a technique adept at disentangling causal interactions from confounding correlations, including those arising from population structure and phylogeny. Through the utilization of this strategy, the study realized two pivotal steps: (i) the identification of non-synonymous SNPs and non-coding variants to generate a comprehensive whole-genome alignment, and (ii) the identification of highly evolutionarily coupled loci, subsequently tested within an exploratory set of 1102 *N. gonorrhoeae* strains using a linear mixed model to evaluate their epistatic associations with MICs for five antibiotics: penicillin, tetracycline, cefixime, ciprofloxacin, and azithromycin. The study’s findings were validated using a separate dataset of 495 *N. gonorrhoeae* strains collected from Canada and England.

Furthermore, it is recognized that a wide variety of observed bacterial phenotypes may be closely linked to the presence or absence of specific genes. These genes can be inherited through descent or acquired through lateral gene transfer. This dynamic results in notable variability in the bacterial pangenome, characterized by differences in the sequences of both the core regions present in each isolate and the accessory regions, which vary in their occurrence [53]. Therefore, Brynildsrud et al. [53] have developed an innovative tool named Scoary, which aims to explore associations between pangenomic genes and observed phenotypes. This approach is termed “pan-GWAS” to distinguish it from traditional GWAS. This method utilizes a permutation test (Fisher’s Exact Test) to analyze clusters of orthologous genes (COGs) and takes into account the phylogenetic structure of the samples to consider the population structure. 

A specific example of this approach is found in the study conducted by Card et al. [25], in which they conducted whole genome sequencing (WGS) and phenotypic susceptibility tests on thirty-four field isolates from the United Kingdom and three control strains. The primary objective of this investigation was to explore pleuromutilin resistance (tiamulin and valnemulin) in *Brachyspira hyodysenteriae*. Employing the Pan-GWAS approach, an innovative pleuromutilin resistance gene, *tva(A)*, was successfully identified. This gene is closely linked to resistance against the antibiotics tiamulin and valnemulin and is responsible for encoding a predicted ABC-F transporter. Furthermore, in vitro cultures of isolates were conducted in the presence of inhibitory or subinhibitory concentrations of tiamulin, demonstrating that the presence of *tva(A)* conferred reduced susceptibility to pleuromutilin. While this reduced susceptibility does not lead to clinical resistance, it does facilitate the development of higher-level resistance through mutations in genes associated with ribosomal functions. These observations underscore the relevance and potential of the “pan-GWAS” approach in deciphering the underlying mechanisms of phenotypic variability in bacteria. 

## 4. Discussion

The studies collected in this review reveal a wide variety of GWAS approaches, including non-phylogenetic, phylogenetic, alignment-free *k*-mers-based, and mixed approaches that significantly contribute to the comprehension of genetic components associated with bacterial phenotypic traits, particularly concerning antibiotic resistance. These findings highlight the evolution that characterizes this field of research, distinguished by significant advances and concurrent challenges. As we deepen our comprehension of genotype–phenotype correlations, novel methodologies surface to surmount the constraints of conventional GWAS methodologies, thereby facilitating a more comprehensive understanding of bacterial resistance mechanisms.

The most commonly used approaches are non-phylogenetic methods that employ a variety of tests and statistical models to investigate genetic associations across a wide spectrum of phenotypic traits. These methods do not necessitate prior knowledge of evolutionary relationships. Among the most frequently utilized approaches are the logistic and linear regression models, which afford the capacity to control for confounding factors and incorporate covariates. This, in turn, facilitates the identification of concrete associations between genetic variants and phenotypes, concurrently diminishing the risk of encountering false positives. However, it remains important to consider their limitations, such as the potential constraint on detecting weak associations attributed to limited sample sizes. Additionally, ensuring robust and meaningful outcomes requires addressing concerns related to false positive control and population structure.

On the other hand, phylogenetic approaches incorporate information from phylogenetic trees to address population structure, thereby enhancing the statistical power to detect genuine associations. This approach is noteworthy for its capacity to elucidate evolutionary relationships among strains by tracing individual mutation events along a phylogenetic tree. In contrast to LMMs, which rely on statistical calculations involving crude ratios [31], these approaches provide a more comprehensive understanding of evolutionary connections. While these methods seem well-suited for bacterial phenotypic traits influenced by homoplastic mutations [39,51], their effectiveness depends on the accuracy of phylogenetic trees, which is especially important in species with recombination, like *S. pneumoniae* and *K. pneumoniae*. Furthermore, their extensive computational requirements can be challenging, particularly when working with large sample sizes [51]. Nevertheless, this innovative approach has successfully identified genetic associations in antibiotic-resistant bacteria [4,23,24,26,28,30,34,36].

An alternative approach is the “Alignment-free *k*-mers-based approach”, which can capture various genetic variations, including SNPs, indels, and gene presence/absence, without requiring prior identification of these variations. Furthermore, it employs LMM and various strategies for population structure control. However, interpreting the results of this approach can be complex due to several factors.

Firstly, a *k*-mer can correspond to multiple regions within the same genome, and a gene can be linked to several *k*-mers, making it complex to assign a specific phenotype to a particular *k*-mer. Secondly, variability in variants and the potential presence of *k*-mers in only specific strains might weaken the association between *k*-mers and the targeted phenotypes compared to other methods that focus on the presence of the gene itself. Thirdly, given that a resistance-associated gene often exists in slightly diverse versions, the *k*-mers for each gene version are only found in a fraction of resistant strains. As a result, these *k*-mers show weaker correlations with resistance than the direct presence of the polymorphic gene [44].

Despite the challenges mentioned previously, substantial progress has been achieved in applying this approach to investigating genetic variants associated with resistance in various bacterial species [8,35,38,40,42,43,44]. *k*-mers-based GWAS approaches are particularly adept at detecting drug resistance associations with mobile genes, as they analyze connections with short DNA fragments derived from the entire genetic content of an organism, extending beyond the analysis of genetic variants solely within the core genome [8,31].

It was observed that the majority of the studies we reviewed are related to conventional GWAS methods based on single locus testing, wherein they evaluate the association between a phenotype and a single variant at a time, iterating this process for all variants across the entire genome [31,51]. These methods frequently overlook crucial factors like concurrent or simultaneous resistance mutations [37], compensatory mutations [16,31], and epistatic interactions [27,49]. However, there have been recent developments in innovative GWAS approaches aimed at enhancing the comprehension of more intricate genotype-phenotype associations. These approaches have given rise to methods that blend statistical techniques or models with phylogenetic information to identify compensatory mutations linked to resistance-related phenotypic traits [16,31]. Furthermore, other strategies integrate linear mixed models alongside data on evolutionary couplings to decipher epistatic interactions (epistatic GWAS), while certain approaches utilize statistical methods with iterative elimination [37].

Over the past decade, the application of the diverse GWAS approaches discussed in this review has demonstrated remarkable success in deepening our understanding of the intricate mechanisms acquired and developed by pathogenic bacteria to counteract the effects of antimicrobial agents. This applies to human pathogens (e.g., *K. pneumoniae*, *Achromobacter* spp., *A. baumannii*, *Burkholderia multivorans*, *Corynebacterium diphtheriae*, *E. coli*, *Mycobacterium abscessus*, *M. tuberculosis*, *N. gonorrhoeae*, *N. meningitidis*, *P. aeruginosa*, *S. aureus*, *Staphylococcus capitis*, *S. pneumoniae*) and pathogens affecting animals, including cattle (e.g., *Streptococcus uberis*, *Mycoplasma bovis*), and pigs (e.g., *Brachyspira hyodysenteriae*). Additionally, have become useful tools for analyzing mechanisms that encompass mutations affecting coding regions [4,8,12,14,19,21,33,34,35,36,38,40,41,42,43,45,47] and non-coding regions of the genome [16,24,28,34,46]. Additionally, the acquisition of resistance genes through horizontal transfer processes or mobile genetic elements has been observed [8,12,18,25,44,48], along with alterations in regulatory elements [22,23,24,29,39,46].

Several studies in this review have demonstrated recent advancements in GWAS research related to antimicrobial resistance. Schubert et al. [27] conducted the first GWAS in *N. gonorrhoeae*, delving into distinct loci and epistatic interactions influencing antimicrobial susceptibility and resistance. Concurrently, Bokma et al. [40] introduced the first GWAS in *M. bovis*, utilizing long-read sequencing and evaluating epidemiological cutoffs (ECOFF). This study revealed new genetic markers linked to acquired resistance, especially to fluoroquinolones and macrolides. Coolen et al. [39] introduced an innovative approach that combines GWAS with homoplasy analysis to investigate how the *E. coli* genome adapts to antibiotics. Furthermore, Sommer et al. [42] emphasized the importance of selecting the appropriate GWAS tool, considering distinct genetic disparities and their distribution in the context of resistance. Conversely, Mortimer et al. [48] conducted the first GWAS in *N. gonorrhoeae* related to penicillin and tetracycline resistance. Their findings revealed promising candidates for molecular diagnostics, pinpointing specific genes and their absences associated with resistance to these antibiotics. Lastly, Boeck et al. [49] introduced an innovative approach that connects multidimensional phenotypes to genotypes by integrating computational structural modeling with GWAS analyses (LMM) and defined epistatic interactions through Correlation-Compressed Direct Docking Analysis (CC-DCA) to create a phenogenomic map for *M. abscessus*, revealing clinically significant phenotypes like virulence and antibiotic resistance. This research provides insights for future therapeutic strategies and predictive analyses for different pathogens. 

Another area of GWAS research that has shown significant progress is the integration of extensive genomic datasets with the quantification of antimicrobial resistance, particularly through MIC measurements. This advancement has enabled the application of GWAS using quantitative resistance phenotypes, allowing for the quantification of the proportion of variation in the resistance phenotype explained by bacterial genetic variation, thereby contributing to an improved understanding of the fundamental mechanisms driving antimicrobial resistance [28,46,47].

In recent years, the application of several machine learning techniques as complementary approaches to GWAS has emerged as a promising strategy. This stems from their capability to address specific limitations and enhance our comprehension of the intricate genetic relationships involved in antibiotic resistance. An illustrative case in point is the study conducted by Mallawarachici et al. [47], wherein a multiple loci model is employed through elastic networks implemented in *pyseer*. Elastic-net regression combines the Least Absolute Shrinkage and Selection Operator (LASSO) with ridge regression, conferring upon it the prowess to tackle high dimensionality and multicollinearity issues. This approach enables an efficacious prediction of the resistance phenotype from genomic data. 

Furthermore, innovative methodologies such as Random Forest have been deployed to categorize strains according to their antibiotic resistance phenotypes. This is achieved by utilizing SNPs that confer resistance and/or genes identified through GWAS, with a focus on bacterial species, including *S. aureus*, *S. pneumoniae*, *A. baumannii*, and *M. tuberculosis* [15,21,30,37]. These findings underscore the suitability of machine learning methods for phenotype prediction, showcasing performance on par with or surpassing that of traditional methods founded on direct associations. Moreover, these approaches could suggest as yet unexplored variants, which could lead to novel discoveries or reveal compensatory mutations and epistasis in a significant way.

In summary, despite that, diverse GWAS approaches have been applied in research aiming to identify genetic variants associated with antimicrobial resistance to understand different bacterial mechanisms for developing resistance to available drugs reported in this review. There is a need to continue working on challenges, including epistasis, compensatory mutations, gene-gene interactions, and environmental factors that may influence the understanding of the mechanisms that contribute to the development of drug resistance. Additionally, another area with considerable potential for exploration is the development of novel computational methods that combine GWAS approaches with machine learning techniques. At the same time, it is necessary to integrate omics data, such as metabolomics, metabolic networks, and protein structural data, into a single analysis. Without a doubt, it could accelerate our understanding of the biological mechanisms of response to drugs, which contributes to the way we address the public health problem, which is resistance to available antibiotics.

Our review was not without some limitations. To focus on the GWAS approaches used to detect genetic variants associated with antibiotic resistance, we excluded those studies that performed GWAS with a different bacterial phenotype. It is possible that studies that used different GWAS approaches that could be applied in the analysis of the antimicrobial-resistant phenotype have not been considered. Given that the included studies showed a wide diversity in the methodologies implemented, with variability in the organisms studied and notable differences in the GWAS approaches used, we chose to refrain from carrying out statistical analyses. Instead, this review focused on describing and comparing the approaches used across studies, highlighting relevant trends and findings.

## 5. Conclusions

This systematic review demonstrates that bacterial GWAS approaches have emerged as fundamental tools for deepening the understanding of the genetic basis of bacterial resistance to antimicrobials across various species. Identification of novel genetic markers associated with antimicrobial resistance represents a crucial advancement that unveils a broad spectrum of opportunities for future research. These findings suggest that GWAS studies possess the potential to propel the development of more precise screening and diagnostic tools, thereby facilitating effective and personalized therapeutic interventions in the battle against antimicrobial resistance.

The diversity of approaches adopted in GWAS studies has proven useful in understanding bacterial resistance mechanisms. Non-phylogenetic, phylogenetic, alignment-free *k*-mers-based, and mixed approaches have played a significant role in elucidating the evolutionary strategies that bacteria have developed to evade the action of antimicrobials. However, it is essential to recognize that there are challenges and open questions in this research field. Further exploration into phenomena such as compensatory mutations and epistasis emerges as promising areas for research aimed at gaining a deeper understanding of the mechanisms underlying resistance evolution.

## Figures and Tables

**Figure 1 microorganisms-11-02866-f001:**
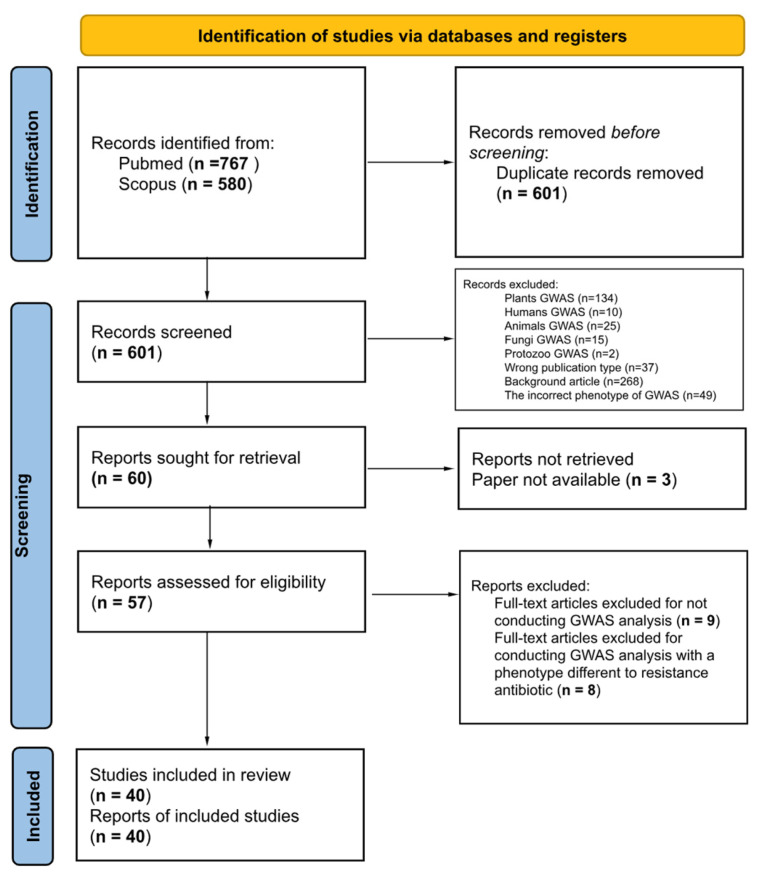
PRISMA flow diagram for the literature review process and selected studies.

**Figure 2 microorganisms-11-02866-f002:**
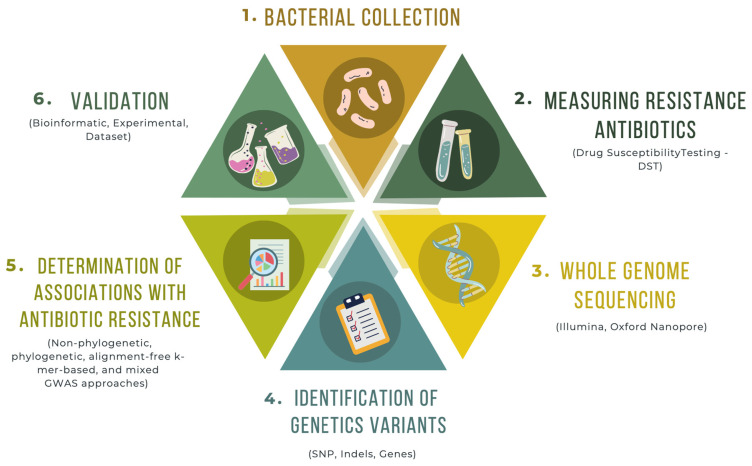
Schematic of a Genome-Wide Association Analysis (GWAS) in Bacteria for Identifying Genetic Variants Associated with Antibiotic Resistance. (1) The process starts with the recollection of a diverse collection of bacterial isolates representative of the population under study. (2) Subsequently, the evaluation of resistance to drugs is conducted through established susceptibility tests. Each isolate in the collection undergoes these tests. (3) Concurrently, next-generation sequencing is employed for whole genome sequencing of each isolate in the collection. (4) From the obtained sequences, genetic variants are identified, including SNPs and indels, as well as the presence or absence of genes within each of the isolates. (5) Following this, analyses are performed to ascertain the association between genetic variants and antibiotic resistance traits. This is executed through the GWAS approach, and/or (6) Genetic variants displaying statistically significant associations with drug resistance can be verified using various methodologies: supplementary datasets of bacterial isolates, genetic, molecular, or functional experiments, along with bioinformatic analyses, can be employed to corroborate these associations.

**Table 1 microorganisms-11-02866-t001:** Eligibility (inclusion and exclusion) criteria used for the screening of titles/abstracts and full texts.

Exclusion Criteria	Inclusion Criteria
Humans GWAS studies	Genome-Wide Association Study (GWAS)
Plants GWAS studies	Bacterial GWAS studies
Animals GWAS studies	Antibiotic resistance phenotype
Fungi GWAS studies	
Protozoon GWAS studies	
Wrong publication type	
Studies not related to GWAS	
GWAS studies with incorrect phenotype	
Studies without GWAS methodology	

**Table 2 microorganisms-11-02866-t002:** Summary of studies included in the systematic review.

Authors	Year	Organism	Samples	Phenotype Resistance	Traits	Genetic Variants	GWAS Software	GWAS Approach	Population Structure Control
[14]	2013	*Escherichia coli*	164	FQ (CIP, GAT, LEV, NOR)	Binary	Genes	Not reported	Arbitrary set arithmetic between the pools based on their phenotypes to enrich for alleles that are linked to a specific phenotypic trait	Not reported
[15]	2014	*Staphylococcus aureus*	75	VAN	Binary and Continuo	SNP	ROADTRIPS (binary phenotypes), QROADTRIPS (continuous phenotype), R (method similar to PhyC)	Regression Models; Phylogenetic	Covariance matrix; Phylogenetic Inference
[16]	2014	*Mycobacterium tuberculosis*	173, 1398	CIP, OFX, EMB, INH, PZA, RIF, STR	Binary	SNP, Genes	GWAMAR	Tree-generalized hypergeometric score (TGH), which incorporates the phylogenetic tree in the analysis, mutual information, odds ratio, hypergeometric test, weighted support	Phylogenetic Inference
[17]	2015	*Escherichia coli*	380	AMP, ATM, CAZ, CFZ, CTT, CRO, SAM, DOX, GEN, SXT, MXF	Binary	Genes	R	Logistic regression model	Genotype matrix obtained with dimension reduction methods (PCA)
[18]	2016	*Acinetobacter baumannii*	120	Carbapenem	Binary	*k*-mers	*bugwas*	Linear Mixed Model that incorporates lineage-specific effects by decomposing the kinship into principal components	Kinship/Relatedness matrix and Genotype matrix obtained with dimension reduction methods (PCA) to test for potential lineage effects
[12]	2016	*Mycobacterium tuberculosis*, *Staphylococcus aureus*, *Escherichia coli*, and *Klebsiella pneumoniae*.	1735, 992, 241, 176	MTB (EMB, INH, PZA, RIF); STA (CIP, ERY, FA, GEN, PEN, MET, TET, RIF, TMP); ECO, KLE (AMP, CFZ, CXM, CRO, CIP, GEN, TOB)	Binary	SNP, Genes, *k*-mers	*bugwas*	Linear Mixed Model that incorporates lineage-specific effects by decomposing the kinship into principal components	Kinship/Relatedness matrix and Genotype matrix obtained with dimension reduction methods (PCA) to test for potential lineage effects
[19]	2016	*Mycobacterium tuberculosis*	127 (40)	RIF, INH, STR, EMB, PAS, ETH, OFX, CAP	Binary and Categorical	SNP	EMMA; PhyC	Linear Mixed Models; Phylogenetic Convergence test	Genotype matrix obtained with dimension reduction methods (PCA); Phylogenetic Inference
[20]	2016	*Mycobacterium tuberculosis*	91	AMC	Continuo	SNP, Genes	EMMA	Phylogenetically controlled Linear mixed model	Kinship matrix and Phylogenetic Inference
[21]	2017	*Streptococcus pneumoniae*	1680	PEN, TMP, CMX, ERY, OFX, CIP	Binary	SNP, Genes	PLINK	Fisher’s exact test	Genetic subpopulations (represented by the sequence clusters; SCs) determined using BAPS
[8]	2018	*Mycobacterium tuberculosis*, *Staphylococcus aureus* and *Pseudomonas aeruginosa*.	5000, 9000, 2500	MTB (RIF, STR, OFX, ETH, XDR-TB); STA (MET, CIP, FA, TMP); PSA (AMK, LEV)	Binary	SNP, *k*-mers	DBGWAS	Compacted De Bruijn graphs (DBG) combined with a Linear Mixed Model	Kinship/Relatedness matrix and Genotype matrix using dimension reduction methods (PCA) to assess potential lineage effects
[22]	2018	*Burkholderia multivorans*	111	ATM, CAZ, AMK, TOB, CIP	Categorical	SNP, Indels	Not report	Fisher’s exact test	Clustering utilizing groups defined by STRUCTURE
[4]	2018	*Neisseria meningitidis*	171	PEN	Binary	SNP, Genes	treeWAS	Phylogenetic	Phylogenetic Inference
[23]	2018	*Mycobacterium tuberculosis*	549	INH	Binary	SNP	phyOverlap	Phylogenetic	Phylogenetic Inference
[24]	2018	*Mycobacterium tuberculosis*	6465	INH, RIF, EMB, ETH, PZA, STR, AMK, KAN, CAP, CIP, MXF, OFX, DCS, PAS, MDR-TB, XDR-TB	Binary	SNP, Genes, small Indels, large deletions	GEMMA; PhyC	Linear Mixed Model; Phylogenetic Convergence test	Kinship matrix; Phylogenetic Inference
[25]	2018	*Brachyspira hyodysenteriae*	37	TIA, VAL	Binary	Genes	Scoary	Pan-GWAS: Fisher’s exact test	Phylogenetic Inference
[26]	2019	*Mycobacterium tuberculosis*	145	ETH	Binary	SNP	phyOverlap	Phylogenetic	Phylogenetic Inference
[27]	2019	*Neisseria gonorrhoeae*	1102	PEN, TET, CFX, CIP, AZM	Continuo	SNP	Python (pylmm)	Linear mixed model; GWAS epistatic: information of evolutionary couplings combined with an adaptation of linear mixed model	Phylogenetic Inference using hierarchically clustering (RhierBAPS)
[28]	2019	*Mycobacterium tuberculosis*	1452	INH, RIF, RFB, EMB, PZA, KAN, AMK, CAP, ETH, STR, MXF	Continuo	SNP, Genes, Indels	GEMMA, treeWAS	Linear Mixed Model; Phylogenetic	Kinship matrix; Phylogenetic Inference
[29]	2020	*Neisseria gonorrhoeae*	4505	AZM, CIP, CRO	Continuo	*k*-mers	*pyseer*	Linear Mixed Model	Phylogenetic matrix and covariable matrix based on the isolate’s country of origin
[30]	2020	*Acinetobacter baurelqii*	84	FEP, CXM, GEN, CAZ, TMP, AZM, CRO, ATM, ERY, PIP, LEV, IPM, CIP	Binary	Genes; *k*-mers; SNP	Python (Scipy); *pyseer;* treeWAS	Mann–Whitney test; Linear model mixed; Phylogenetic	NA; Kinship matrix; Phylogenetic Inference
[31]	2020	*Mycobacterium tuberculosis*	600	INH, RIF, EMB, STR, PZA, KAN, CIP, ETH, PAS	Binary	SNP	ECAT	Adaptation of the Linear Mixed Model to integrate homoplasy information while accounting for confounding factors	Kinship/Relatedness matrix
[32]	2020	*Neisseria gonorrhoeae*	4505	AZM	Continuo	*k*-mers	*pyseer*	Linear Mixed Model	Phylogenetic matrix
[33]	2020	*Streptococcus uberis*	265	OXA	Continuo	*k*-mers	SEER	Linear regression	Distance matrix applied multidimensional scaling (MDS)
[34]	2020	*Mycobacterium tuberculosis*	549 (1635, 1365)	INH	Binary and Continuo	SNP	phyOverlap	Phylogenetic	Phylogenetic Inference
[35]	2020	*Staphylococcus capitis*	162	VAN	Continuo	*k*-mers	DBGWAS	Compacted De Bruijn graphs (DBG) combined with a Linear Mixed Model	Kinship/Relatedness matrix and Genotype matrix using dimension reduction methods (PCA) to assess potential lineage effects
[36]	2020	*Corynebacterium diphtheriae*	247	PEN, AMX, OXA, CTX, IMP, AZM, CLR, ERY, SPR, CLI, PRT, GEN, KAN, SUL, TMP, STX, RIF, TET, CIP	Binary	SNP, Genes	treeWAS	Phylogenetic	Phylogenetic Inference
[37]	2020	*Mycobacterium tuberculosis*	3574	INH, RIF, EMB, PZA, STR	Categorical	SNP	Hungry SNPos algorithm (HHS)	Simple scoring heuristic combined with iterative ‘cannibalism’	Iterative ‘cannibalistic’ elimination algorithm
[38]	2021	*Achromobacter* spp.	92	AMC, CAZ, CHL, CST, IPM, MEM, TZP, SMZ, TGC, SXT	Binary	*k*-mers (*unitig*)	DBGWAS	Compacted De Bruijn graphs (DBG) combined with a Linear Mixed Model	Kinship/Relatedness matrix and Genotype matrix using dimension reduction methods (PCA) to assess potential lineage effects
[39]	2021	*Escherichia coli*	172	CTX	Binary	SNP	R; *pyseer*	Fisher’s exact test; Linear Mixed Model	NA; Phylogenetic Inference
[40]	2021	*Mycoplasma bovis*	95	OTC, DOX, TIL, TYL, GAM, FLO, GEN, ENRO, TIA	Categorical	*k*-mers	DBGWAS	Compacted De Bruijn graphs (DBG) combined with a Linear Mixed Model	Kinship/Relatedness matrix and Genotype matrix using dimension reduction methods (PCA) to assess potential lineage effects
[41]	2021	*Escherichia coli*	1178	NAL, NOR, CIP, LEV	Continuo	SNP, Indels	SEER	Fixed Effects Model	Phylogenetic matrix with multidimensional scaling (MDS)
[42]	2021	*Staphylococcus aureus*	283	MRLM	Binary	Genes, *k*-mers, SNP	Scoary; DBGWAS; PLINK	Pan-GWAS: Fisher’s exact test, Linear Mixed Model, and Fisher’s exact test	Phylogenetic Inference; Kinship/Relatedness matrix and Genotype matrix using dimension reduction methods (PCA) to assess potential lineage effects; Genetic relatedness into models are delineation of strain clusters
[43]	2022	*Achromobacter* spp.	54	SXT, TGC, SSS, IPM, TZP, MEM.	Categorical	*k*-mers	DBGWAS	Compacted De Bruijn graphs (DBG) combined with a Linear Mixed Model	Kinship/Relatedness matrix and Genotype matrix using dimension reduction methods (PCA) to assess potential lineage effects
[44]	2022	*Pseudomona aeruginosa*	280	AMK	Binary	*k*-mers	CALDERA	Cochran–Mantel–Haenszel (CMH) test	Not reported
[45]	2022	*Mycobacterium tuberculosis*	1184, 1163, 1159	AMK, CAP, KAN	Binary	SNP, Genes	Python script (https://gitlab.com/LPCDRP/gwa)	Not reported	Not reported
[46]	2022	*Mycobacterium tuberculosis*	10,228	INH, RIF, EMB, AMK, ETH, KAN, LEV, MXF, RIB, BDQ, CLF, DLM, LNZ	Continuo	*k*-mers	GEMMA	Linear Mixed Model	Kinship/Relatedness matrix
[47]	2022	*Streptococcus pneumoniae*	1612	PEN, CRO	Continuo	Genes	lme4qtl (R package), *pyseer*, treeWAS	Linear Mixed Model/Generalized Least Squares regression (frequency-based allele coding); Linear Mixed Model; Phylogenetic	NA; Kinship matrix; Phylogenetic inference
[48]	2022	*Neisseria gonorrhoeae*	9673	PEN, TET	Binary	*k*-mers	*pyseer*	Linear Mixed Model	Kinship/relatedness matrix and Covariable matrix based on the isolate dataset’s origin, country of origin, and presence of plasmid-encoded resistance determinants (blaTEM, t*etM*)
[49]	2022	*Mycobacterium abscessus*	331	AMK, FOX, CLR, LNZ, CFZ	Continuo	SNP, Indels	GEMMA, *bugwas*	Linear Mixed Model: Linear Mixed Model that incorporates lineage-specific effects by decomposing the kinship into principal components	Kinship/Relatedness matrix; Kinship/Relatedness matrix and Genotype matrix obtained with dimension reduction methods (PCA) to test for potential lineage effects
[50]	2023	*Mycobacterium tuberculosis*	2773	MDR, poly-drug resistant, pre-XDR, XDR.	Binary	SNP	GAPIT	Compressed Mixed Linear Model	Kinship matrix and Genotype matrix were obtained with dimension reduction methods (PCA)

Abbreviations: FQ—Fluoroquinolones; CIP—ciprofloxacin; GAT—gatifloxacin; LEV—levofloxacin; NOR—norfloxacin; VAN—vancomycin; OFX—ofloxacin; EMB—ethambutol; INH—isoniazid; PZA—pyrazinamide; RIF—rifampin; STR—streptomycin; AMP—ampicillin; ATM—aztreonam; CAZ—ceftazidime; CFZ—cefazolin; CTT—cefotetan; PIP—piperacillin; SAM—ampicillin-sulbactam; GEN—gentamicin, DOX—doxycycline; SXT—trimethoprim-sulfamethoxazole; CTX—cefotaxime; MTB—*Mycobacterium tuberculosis*; STA—*Staphylococcus aureus*; ERY—Erythromycin; FA—fusidic acid; PEN—Penicillin; MET—Methicillin; TMP—Trimethoprim; ECO—*Escherichia coli*; KLE—*Klebsiella pneumoniae*; CXM—Cefuroxime; CRO—Ceftriaxone; TOB—Tobramycin; PAS—para-aminosalicylic acid; ETH—Ethionamide; CAP—Capreomycin; AMC—amoxicillin-clavulanic acid; CMX—cotrimoxazole; XDR-TB—Extensively drug-resistant tuberculosis; PSA—*Pseudomonas aeruginosa*; AMK—amikacin; CIP—ciprofloxacin; MXF—moxifloxacin; DCS—D-cycloserine; MDR-TB—Multidrug-resistant tuberculosis; TIA—tiamulin; VAL—valnemulin; RFB—rifabutin; TET—tetracycline; CFX—cefixime; AZM—azithromycin; FEP—Cefepime; IPM—Imipenem; OXA—oxacillin; CLR—clarithromycin; SPR—spiramycin; CLI—clindamycin; PRT—pristinamycin; KAN—kanamycin; SUL—sulfonamide; CHL—Chloramphenicol; CST—Colistin; MEM—Meropenem; TZP—Piperacillin-Tazobactam; SMZ—Sulfamethizole; TGC—Tigecycline; OTC—oxytetracycline; DOX—doxycycline; TIL—Tilmicosin; TYL—tylosin tartrate; FLO—florfenicol; GAM—gamithromycin; NAL—nalidixic acid; NOR—Norfloxcin; MRLM—meticillin resistant lacking mec; SSS—Sulfonamides; BDQ—bedaquiline; CLF—clofazimine; DLM—delamanid; LNZ—linezolid; FOX—Cefoxitin; pre-XDR—Preextensively drug-resistant tuberculosis; SNP—Single-Nucleotide Polymorphisms; *k*-mers—DNA words of length *k*; PCA—Principal Components Analysis.

## Data Availability

Data are contained within the article or Appendix A.

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
