# Peer review of "Genome-Wide Association Studies (GWAS) Approaches for the Detection of Genetic Variants Associated with Antibiotic Resistance: A Systematic Review"

_microorganisms, 2023, doi:10.3390/microorganisms11122866_

Round 1

Reviewer 1 Report

Comments and Suggestions for Authors

The main question addressed by the systematic review: Genome-wide association studies (GWAS) approaches for the detection of genetic variants associated with antibiotic resistance. This topic very important, because identification of novel genetic markers associated with antimicrobial resistance represents a crucial advancement that unveils a broad spectrum of opportunities for future research.

The references appropriate. The number of references (53) no meet the requirements of the journal for articles, in addition, the number of sources five years ago (2018-2023) is 56,4% (37), which is enough. But the number of reference for systematic review about  80-100.

Line 306: Bugwas has demonstrated….

Line 618: A. baumannii, and M. tubercu

Figures and table are illustrative and informative.

In Materials and Methods no information about statistics methods. 

Good luck!

Author Response

Dear Reviewer,

We sincerely appreciate your valuable comments on our manuscript and the time spent reviewing it. Your words strengthen our dedication to delving deeper into this field and underscore its significance in the area of microbiology and antibiotic resistance research.

Point-by-point response to Comments and Suggestions for Authors
Comments 1: The references appropriate. The number of references (53) no meet the requirements of the journal for articles, in addition, the number of sources five years ago (2018-2023) is 56,4% (37), which is enough. But the number of reference for systematic review about  80-100.
Response 1:  We have reviewed the instructions for the author of the journal Microorganisms(https://www.mdpi.com/journal/microorganisms/instructions#preparation) and found no specific requirements limiting the number of bibliographic references in systematic review manuscripts. Furthermore, upon examining recent systematic reviews published in the Microorganisms journal,  we have observed a variability in the number of bibliographic references cited in these articles. Some of them included 27, 50, 73, and even 125 references.  This suggests that the number of references in systematic reviews may be influenced by the specific research field, the availability of relevant literature, and the presence of studies that meet the inclusion criteria. Based on this insight, we believe that including 53 references in our manuscript is sufficient to substantiate our systematic review.
Comments 2:  Line 306: Bugwas has demonstrated….
Line 618: … A. baumannii, and M. tubercu
Response 2:  Thank you for pointing this out. We have incorporated the suggested changes in the manuscript, specifically in relation to Lines 306 and 618. You will find these modifications highlighted in red within the document.
Comments 3:  In Materials and Methods no information about statistics methods. 
Response 3:  We wish to clarify aspects related to the orientation of our study. Our review focused on performing a narrative synthesis of GWAS approaches applied to bacteria, specifically for the identification of genetic variants linked to antibiotic resistance. During the process of reviewing the literature that met the inclusion criteria, we noted significant diversity in the study designs, populations studied, and interventions evaluated in the included studies. This variability precluded robust quantitative analysis. Therefore, we opted for a qualitative synthesis, summarizing and comparing the approaches in the included studies without formal statistical analyses, which we deemed more suitable due to significant variations among the studies. In response to your comment on the Materials and Methods section, we are willing to highlight this situation as a limitation of our study, emphasizing the qualitative nature of our approach and the reason behind the omission of formal statistical analyses. From lines 641 to 645, we have included the requested clarifications. "Given that the included studies showed a wide diversity in the methodologies implemented, with variability in the organisms studied and notable differences in the GWAS approaches used, we chose to refrain from carrying out statistical analyses. Instead, this review focused on describing and comparing the approaches used across studies, highlighting relevant trends and findings."  You will find these modifications highlighted in red within the document.

Reviewer 2 Report

Comments and Suggestions for Authors

In the manuscript entitled 'Genome-wide association studies (GWAS) approaches for the detection of genetic variants associated with antibiotic resistance: a systematic review', the authors conduct a systematic review of the different GWAS approaches.

I find the manuscript well structured and well written. The introduction provides sufficient background and includes the purpose of this systematic review, which is to describe the bacterial GWAS approaches used to identify genetic variants associated with antimicrobial resistance. The materials and methods clearly describe the source selection process, as well as the eligibility criteria and the search strategy. The results are well presented and discussed and support the conclusions drawn by the authors.

Author Response

Dear Reviewer,

We are grateful for your valuable comments on our manuscript and the time you spent reviewing it. Your comments are an incentive for our team, and we are pleased to know that you find the manuscript well-structured and clearly written.

Sincerely,

Jeanneth Mosquera-Rendón

Reviewer 3 Report

Comments and Suggestions for Authors

The rapid spread of antibiotic resistance among pathogenic bacteria poses an increasing threat to humans. As a result, research efforts to understand the mechanisms of AMR are being intensified. This also involves an increasing amount of data for analysis. The present work is a valuable overview of genome-wide association studies (GWAS) which have emerged as a powerful tool to identify genetic variants associated with AMR. I have no significant comments on the manuscript. In my opinion it meets the criteria of the Microorganisms and is worth publishing.

Author Response

Dear Reviewer,

We sincerely appreciate your valuable comments on our manuscript and the time you spent reviewing it. Your opinion that the manuscript meets the Microorganisms criteria and deserves publication is a great honor for us. We are truly grateful for your support and the positive evaluation you've provided, which motivates us to continue contributing to this area of research.

Sincerely,

Jeanneth Mosquera-Rendón